# Microbiota Alterations and Their Association with Oncogenomic Changes in Pancreatic Cancer Patients

**DOI:** 10.3390/ijms222312978

**Published:** 2021-11-30

**Authors:** Heidelinde Sammallahti, Arto Kokkola, Sama Rezasoltani, Reza Ghanbari, Hamid Asadzadeh Aghdaei, Sakari Knuutila, Pauli Puolakkainen, Virinder Kaur Sarhadi

**Affiliations:** 1Department of Pathology, Faculty of Medicine, University of Helsinki, 00014 Helsinki, Finland; heidelinde.sammallahti@helsinki.fi; 2Department of Surgery, Abdominal Center, Helsinki University Hospital and University of Helsinki, 00290 Helsinki, Finland; arto.kokkola@hus.fi (A.K.); pauli.puolakkainen@hus.fi (P.P.); 3Foodborne and Waterborne Diseases Research Center, Research Institute for Gastroenterology and Liver Diseases, Shahid Beheshti University of Medical Sciences, Tehran P.O. Box 1985717411, Iran; samasoltani70@gmail.com; 4Digestive Oncology Research Center, Digestive Diseases Research Institute, Tehran University of Medical Science, Tehran P.O. Box 1411713135, Iran; r.ghanbari98@gmail.com; 5Basic and Molecular Epidemiology of Gastrointestinal Disorders Research Center, Research Institute for Gastroenterology and Liver Diseases, Shahid Beheshti University of Medical Sciences, Tehran P.O. Box 1985717411, Iran; hamid.assadzadeh@gmail.com; 6Department of Oral and Maxillofacial Diseases, Helsinki University Hospital and University of Helsinki, 00290 Helsinki, Finland; virinder.sarhadi@helsinki.fi

**Keywords:** pancreatic cancer, pancreatic ductal adenocarcinoma, microbiota, dysbiosis, oncogenomics, mutations, inflammation, drug response, bacterial metabolites

## Abstract

Pancreatic cancer (PC) is an aggressive disease with a high mortality and poor prognosis. The human microbiome is a key factor in many malignancies, having the ability to alter host metabolism and immune responses and participate in tumorigenesis. Gut microbes have an influence on physiological functions of the healthy pancreas and are themselves controlled by pancreatic secretions. An altered oral microbiota may colonize the pancreas and cause local inflammation by the action of its metabolites, which may lead to carcinogenesis. The mechanisms behind dysbiosis and PC development are not completely clear. Herein, we review the complex interactions between PC tumorigenesis and the microbiota, and especially the question, whether and how an altered microbiota induces oncogenomic changes, or vice versa, whether cancer mutations have an impact on microbiota composition. In addition, the role of the microbiota in drug efficacy in PC chemo- and immunotherapies is discussed. Possible future scenarios are the intentional manipulation of the gut microbiota in combination with therapy or the utilization of microbial profiles for the noninvasive screening and monitoring of PC.

## 1. Introduction

Pancreatic cancer (PC) is an aggressive malignancy with a high mortality and poor prognosis. Even though the world-wide incidence is relatively low (2.3% of new cancer cases per year), PC accounts for 4.7% of cancer-related deaths [1]. In the US and Europe, with a 3% incidence, PC accounts for 7–9% of cancer-related deaths, which represents the third and fourth most common cause of neoplastic deaths, respectively [2,3]. The incidence of PC in Central Africa and Central Asia is the lowest worldwide; however, some developing countries such as Iran, have similar incidences to Western countries [4,5].

PC often has no specific symptoms and is detected at a locally advanced or metastatic stage when it is too late for a curative treatment. In addition to this, PC tends to develop resistance to chemotherapy, which leads to recurrence and death in most cases [6]. Although the five-year survival rate in the US has improved slightly (2–3% to 10%) in the past decades [7], due to improved treatment regimens and more efficient screening, it is still very low. The mortality rate of PC has remained more or less stagnant or even increased over the past decades in the US and Europe [8] and PC is projected to become the second most common cause of cancer-related death in the US by 2030 [9].

Histologically, PCs can be grouped into exocrine and neuroendocrine tumors. About 95% of PCs are exocrine tumors, which originate from pancreatic exocrine cells composed of acinar and ductal cells. Pancreatic ductal adenocarcinoma (PDAC) is the most prevalent class of PC accounting for over 90% of all pancreatic malignancies [10]. PDAC most commonly arises from pancreatic intraepithelial neoplasms (PanINs), that gradually develop into malignancy by accumulating somatic mutations.

Risk factors associated with the development of PCs include advanced age (median 71 years), gender (males have a slightly higher incidence than females), a family history of PC, cigarette smoking, heavy alcohol consumption, chronic pancreatitis, obesity, type 2 diabetes mellitus and diet [11,12]. About 5%–10% of PCs are associated with inherited risk factors that include familial pancreatic cancer, familial adenomatous polyposis, familial atypical multiple mole melanoma, and Peutz–Jeghers syndrome [13,14].

The investigation of cytogenetic, genetic, and epigenetic changes in PC has led to a better understanding of tumorigenesis and has opened up new possibilities of early diagnosis and disease monitoring in cancer. Mutations in *KRAS* are the most frequent genetic changes, followed by mutations in tumor suppressors *TP53*, *SMAD4* and *CDKN2A* [15] that cause changes in multiple molecular pathways, resulting in cancer development. However, the tumorigenesis of PC has not been fully uncovered and no efficient treatment has been found.

Over the past decade, the human microbiome has become a hot topic in biomedical research, as the development of high throughput methods has facilitated the simultaneous detection of hundreds of different species of microorganisms in a single sample [16]. The gut microbiome is an essential factor in several physiological processes, including host energy metabolism, an inflammatory state, gut permeability, and peptide hormone secretion [17]. In cancer research, the significance of microorganisms in the development of certain malignancies, like *Helicobacter pylori* in gastric cancer and human papillomavirus (HPV) in cervical cancer, is widely recognized [18]. It has become clear that microorganisms can (1) participate in tumorigenesis, (2) influence the host immune response, and (3) alter microbial and host metabolism [19]. Apart from these tumorigenic qualities, microbes may be exploited as anticancer agents by their ability to infect and survive within the human body. They could be used to deliver bioactive molecules like toxins to tumor cells, thus killing them. *Clostridium* and *Salmonella*, for example, have been shown to survive within tumors [20].

Due to their physical proximity, malignancies of the gastrointestinal tract (GIT) are prone to being influenced by the gut microbiota. In our previous studies, we were able to show associations between different kinds of GIT neoplasms and certain gut microbiota profiles [21,22]. In the case of colorectal cancer (CRC), for example, studies on the gut microbiome have led to a better understanding of CRC development and the identification of microbiome biomarkers, and they have provided insight into the influence of microbiota on cancer therapy [23]. The role of the microbiome in PC, however, has not yet been studied that well, and there are still controversies regarding its impact on PC tumorigenesis, progression, and drug resistance [24]. 

In this review, we elucidate the latest findings of microbiota research in PC. Special focus is laid on possible connections between the microbiome, oncogenic changes, and PC. Identifying such associations will help in understanding the role of microorganisms in the tumorigenesis mechanisms of PC, and could have an impact on early diagnosis, more effective treatment, and the improvement of survival rates in PC patients.

## 2. Genomic Characteristics of Pancreatic Cancer

PC is marked by a highly complex karyotype with multiple structural and numerical chromosome abnormalities, including recurrent aberrations involving known oncogenes and tumor suppressors [25]. Although the mutational landscape of PC is characterized by numerous somatic copy number alterations (CNAs) and gene mutations, only four genes play a major role and are affected in most sporadic PDAC cases. These include the oncogene *KRAS* and the tumor suppressors *TP53*, *SMAD4* and *CDKN2A* [15]. One of the earliest events in tumorigenesis that is detectable in almost all PDAC precursor lesions is *KRAS* oncogene activation, which might be considered as the molecular feature of this malignancy [26]. The Cancer Genome Atlas Network (TCGA) published an integrated genomic characterization of PDAC patients, utilizing a multiplatform approach that included genomics, transcriptomics, and proteomics [15]. Mutated *KRAS* was detected in 93% of PDAC patients, with the G12D allele in 41%, the G12V allele in 27%, and the G12R allele in 19% of cases. In 5% of the patients, multiple distinct *KRAS* mutations were detected, while in 7% of the cases with wild-type *KRAS*, mutations in other RAS pathway genes, or alternative oncogenic drivers were found. The frequencies of *TP53*, *SMAD4,* and *CDKN2A* mutations were 72%, 32%, and 30% of the cases, respectively. Less frequently mutated genes were the oncogenes *GATA6*, *GNAS*, *AKT2*, *FGFR1*, *MYC*, *BRAF* and *MDM2*, chromatin modification genes *ARID1A*, *PBRM1*, *MLL3* and *MLL4*, tumor suppressors *PTEN* and DNA damage-repair genes *BRCA1, BRCA2*, *ATM* and *PALB2* [15].

## 3. The Microbiome in Pancreatic Health

The pancreas plays an important role in metabolism through the secretions of its exocrine and endocrine glands. While the exocrine gland controls digestion by producing pancreatic juice (digestive enzymes and sodium bicarbonate), the endocrine gland secretes islet peptide hormones to maintain glucose homeostasis [27]. A healthy pancreas is pivotal in controlling the gut microbiota and reciprocally the gut microbiota has a key impact on pancreatic function [28]. Studies on mouse models have shown that antimicrobial peptides secreted by the pancreas control the composition and diversity of the gut microbiota, and, as a consequence, protect against inflammation. The cathelicidin-related antimicrobial peptide (CRAMP), produced by pancreatic endocrine β-cells, destroys unwanted intestinal microbes by permeabilizing the bacterial membrane [29]. A preclinical study demonstrated that the absence of CRAMP due to impaired exocytosis leads to an alteration in the gut microbiota and bacterial overgrowth in pancreatic cells, and further, to intestinal inflammation and death. Supplementation of CRAMP, however, prevented this bacterial overgrowth and inflammation, which confirms the influence of the pancreas on the gut microbiota [30].

On the other hand, certain metabolites produced by gut bacteria affect pancreatic function. In a study on mice, butyrate, a short-chain fatty acid (SCFA) produced by intestinal bacteria, has been shown to induce the expression of CRAMP in pancreatic β-cells [31]. Likewise, acetate, another SCFA metabolite of gut bacteria, induced insulin secretion in rats via a microbiome–brain β-cell axis [32]. This mutual interaction between the gut microbiota and the pancreas is thus important both in health and disease. Bacterial imbalance or dysbiosis may lead to a dysfunctional pancreas and result in disease, and reciprocally pancreatic disease may cause intestinal dysbiosis (Figure 1).

The gut microbiota is, nevertheless, not the only potential influencer of pancreatic health. Contrary to previous assumptions, the pancreas itself is not sterile and has its own microbiota. The presence of bacterial DNA in pancreatic tissue has been reported in 76% of PDAC patients and 15% of healthy individuals [33]. Microbes are thought to migrate from the duodenum to the pancreas through the pancreatic duct. A comparison of the microbiota from different gastrointestinal sites has shown an overlap of the pancreatic and duodenal microbiome both in PC patients and in healthy controls, affirming that pancreatic bacteria may migrate from the intestine. The pancreatic microbiota have been reported to be very diverse and include certain taxa typically detected in the oral cavity. Moreover, the pancreatic bacterial diversity in PC patients has been found to vary significantly from that of healthy controls [34]. The presence of bacterial DNA in pancreatic tissue samples was also confirmed by Thomas et al., but they did not find significant differences in genus richness or diversity between cancerous and noncancerous tissue [35].

Pushalkar et al. have demonstrated migration of oral fluorescently labeled *Enterococcus faecalis* to the pancreas via the intestine, with a higher level of migration in PDAC mice compared to noncancerous mice [17]. Interestingly, in an experiment with germ-free mice, the pancreas was not colonized by bacteria under normal physiological conditions [35]. Besides the pancreatic duct, alternative ways of colonization of the pancreas have been suggested, including oral, mesenteric venous drainage, and mesenteric lymphatic drainage routes [36]. These partly contradictory findings illustrate that the questions regarding the origin and significance of the intrapancreatic microbiome are not fully resolved. Further studies on larger cohorts are needed for clarification.

## 4. Microbiota Alterations in PC

In addition to influencing the physiological functions of the pancreas, microbial dysbiosis can also enhance inflammation and affect tumorigenic processes, such as cellular proliferation, invasion, metastasis, angiogenesis, and immune modulation [37]. Besides genetic and environmental factors, the oncogenic microbiome, or oncobiome, is one of the regulators of the hallmarks of cancer [38,39]. In addition to the local tumorigenic effects, alterations in microbiota may also exert hormone-like, long-distance effects on different organs [40]. The following subsections provide an overview of microbial alterations in different parts of the gastrointestinal tract and their possible significance in PC. A graphic summary of these alterations is shown in Figure 2.

### 4.1. Oral Microbiota and PC

Several studies have shown an association of oral bacterial dysbiosis with PC tumorigenesis [41,42,43]. The spread of oral microbes to the pancreas via translocation or dissemination has been discussed in the previous section [44,45]. Periodontal disease, a condition linked to alterations in oral bacteria, has been related to an increased risk of PC [46]. Poor oral health, pathogenic oral flora, periodontal disease, and tooth loss are well-established independent risk factors for PDAC [41].

Unique oral microbiota profiles have been associated with PDAC in several studies. PC was associated with significantly increased abundances of the oral bacteria *Porphyromonas gingivalis* [47,48,49], *Fusobacterium* [49], *Graniculatella adiacens* [50] and *Leptotrichia* [48,49] and significantly decreased abundances of *Neisseria elongata* and *Streptococcus mitis* [50], amongst others. Combinations of certain microbes, like *N. elongata* and *S. mitis*, or the increased ratio of *Leptotrichia* to *Porphyromonas* were found to significantly differentiate PC patients from healthy controls and were thus suggested as potential predictive biomarkers for the early detection of PC, especially since saliva sampling is noninvasive and very easy to arrange [48,50].

### 4.2. Pancreatic Microbiota and PC

Preclinical models have shown that the microbial abundance in pancreatic tumor tissue is up to a thousand times higher than in healthy pancreatic tissue [17,51]. The PDAC microbiota has been profiled in several studies, with partly similar, and partly differing results [17,34,42,52,53,54,55]. The first pathogen to be detected in pancreatic tumor tissue and associated with PC was *H. pylori* [56]. *H.pylori* DNA was reported in the pancreatic tissue of 75% of PDAC patients, in 60% of patients with chronic pancreatitis, but in none of the healthy controls [57]. In addition, the abundance of *Fusobacterium* spp., an oral pathogen, was found to be significantly increased in PDAC tissue compared to controls and was associated with a worse prognosis [34,42]. *Lactobacillus*, on the other hand, was more common in healthy controls than in PDAC patients [34]. An increased abundance of Firmicutes and Proteobacteria, which are also the most prominent phyla of the healthy gut, has been observed in several studies in PC tissue as compared to healthy pancreatic tissue [36,55,56]. An enrichment of the fungal microbe *Malassezia* spp. in PDAC tumor tissue has also been reported [52]. In addition to that, pancreatic cyst fluid was found to contain its own unique microbiome [58]. An increased abundance of the oral bacteria *Fusobacterium nucleatum* and *Granulicatella adiacens* was detected in the pancreatic cyst fluid of IPMNs compared to non-IPMN pancreatic cystic neoplasia. Since IPMNs can develop into invasive PC, the results point to the possible pathogenicity of these species and underscores the likelihood of bacterial colonization from the oral cavity [59].

### 4.3. The Intestinal Microbiota and PC

As the pancreas is connected to the intestine through the pancreatic duct, it is obvious that the gut microbiota can influence the pancreas and vice versa [36]. To investigate the relationship between gut microbial dysbiosis and PC, the intestinal tissue microbiota, as well as the fecal microbiota of PC patients have been studied. Clear associations of a unique gut microbiome profile with PDAC were shown in several studies [56,60,61,62,63], and a significant decrease in the microbial alpha diversity was observed in PDAC cancer patients compared to healthy controls [56,61,64].

Summarizing the main findings, *H. pylori* infection in the upper gastrointestinal tract has been associated with an increased risk of developing PDAC [65,66]. *H.* pylori is believed to have an impact on carcinogenesis by promoting cell proliferation [67]. Significantly increased abundances of bacteria belonging to the phyla Bacteroidetes, Firmicutes [61,68], Proteobacteria, Actinobacteria, Fusobacteria, Verrucomicrobia [34,36,69], the genera *Porphyromonas, Prevotella, Bifidobacterium* [34] and *Synergistetes,* as well as the archaeal phylum *Euryarchaeota* [17], have been reported in PC in comparison to healthy controls. Other changes in the gut microbiota reported in PC, were decreased abundances of *Firmicutes*, *Proteobacteria* [61] and *Lactobacillus* [34]. Ren et al. noted an increase in the abundance of potentially pathogenic lipopolysaccharide (LPS)-producing bacteria and a decrease in the abundance of beneficial probiotics and butyrate-producing bacteria in PC patients [61]. Results of different studies are partly similar, but partly contradictory, indicating that larger studies will be needed for establishing a clearer profile of the PDAC gut microbiome.

## 5. Microbiota in Pancreatic Inflammation, Oncogenesis and Tumor Immunity

Inflammation of the pancreas plays a key role in the development of pancreatic cancer. One cause of pancreatic inflammation is a dysbiotic oral, gastric, or intestinal microbiota that can cause an overgrowth of harmful bacteria. This can lead to epithelial barrier breaches and the migration of bacteria to the pancreas. Continual colonization of the pancreas by dysbiotic bacteria results in persistent inflammation and promotes cancer development [24]. 

Microbial products or metabolites support tumor growth by maintaining inflammation and by immune modulation [37]. Bacterial products such as LPS, SCFAs, lipoproteins, lipopeptides, as well as CpG DNA and single- or double-stranded DNA, can induce immune suppression by binding to pattern recognition receptors (PPR) and by activating Toll-like receptors (TLR). This promotes tumor growth by immune evasion, especially during early carcinogenesis [70]. LPS-induced TLR signaling is also thought to help in maintaining inflammation in PC [37]. 

Inflammation can also contribute to PC development through its oncogenic effect. Chronic inflammation in pancreatic tissue can trigger KRAS oncogenic mutation in insulin-positive endocrine cells and induce the differentiation of epithelial cells, resulting in PDAC [69]. KRAS is mutated in 93% of PC cases [15] and despite being a common mutation, the activation of KRAS can still require hyperstimulation from LPS-driven inflammation [61]. The activated KRAS can further advance carcinogenesis by activating the nuclear factor kappa B (NF-κB) pathway [71]. 

A distinct tumor microbial profile (*Pseudoxanthomonas/Streptomyces/Saccharopolyspora/Bacillus clausii*) has been linked with longer survival of PC patients. Moreover, a higher diversity of tumor microbiota was found associated with higher tumor infiltration of T-cells. Notably, the tumor bacterial profile of long-term survivors was associated with higher infiltrating CD8^+^ T-cells expressing granzyme B and better cytotoxic T-cell responses [53].

## 6. The Microbiota and Drug Response in PC

PC patients are often treated with gemcitabine-based chemotherapy [72] and they frequently develop chemoresistance and reduced drug sensitivity [73]. It has been shown that the microbiota plays an important role in the therapeutic efficacy in PC [74]. In CRC mouse models, Geller and colleagues observed that the enzyme cytidine deaminase, produced by Gammaproteobacteria (especially *Mycoplasma hyorhinis*), metabolizes gemcitabine into its inactive form. Gemcitabine resistance was induced by intratumor Gammaproteobacteria and abrogated by antibiotic treatment [33]. Moreover, they detected bacterial DNA, mostly belonging to Gammaproteobacteria in 76% of the tumors of PDAC patients. Therefore, they suggested that antibiotics could be coadministered with gemcitabine therapy to prevent the development of drug resistance [33]. Similarly, *Fusobacterium nucleatum* was shown to promote chemoresistance in CRC [75]. However, the gut microbiota can also have positive effects in chemotherapy. For example, *Lactobacillus plantarum* culture supernatant had a favorable influence on the treatment of colorectal cancer cells with 5-fluorouracil by increasing its chemosensitivity [76]. Likewise, *Enterococcus hirae* and *Barnesiella intestinihominis* improved the therapeutic efficacy of cyclophosphamide by facilitating immunomodulatory effects [77].

Immunotherapeutic approaches in PC that are currently under investigation include immune checkpoint inhibitors (ICIs), vaccine therapy, adoptive cell transfer, myeloid-targeted therapy, immune agonist therapy and combinations with chemoradiotherapy or other molecularly targeted agents [78,79]. Bacteria can exert both positive or negative influences on the immune response and immunotherapies. For example, *Bacteroidetes* spp. were shown to activate Th1 immune responses, and *Listeria monocytogenes* changed tumor-associated macrophages from the immunosuppressive M2 phenotype to the antitumor M1 phenotype [74]. The immune response in cancer therapy was improved by the inhibition of regulatory T cells (Tregs) through *Bifidobacterium adolescentis*, *Enterococcus faecium, Collinsella aerofaciens* and *Parabacteroides merdae* [74]. The gut microbiota has been shown to increase the efficacy of blockade therapy of programmed cell death 1 (PD-1) protein and its ligand, programmed cell death ligand 1 (PD-L1) [80]. On the contrary, the anticancer immune response increased and the tumor burden was reduced by depletion of the gut microbiota through oral gavage antibiotics treatment in a mouse model of PC [81].

The use of gut microbes in combination with immunotherapies has been suggested for the future [74]. However, their mechanisms in enhancing or attenuating the efficacy of immunotherapies need to be identified. Through fecal microbiota transplant (FMT) or supplementation with certain prebiotics, probiotics, or antibiotics, the gut microbial composition could be manipulated to enhance host anticancer immunity and combat drug resistance [80]. Moreover, the gut microbiota could be used as a biomarker for drug efficacy, treatment response and drug side effects [74].

## 7. Associations of Bacterial Metabolites with Carcinogenic and Oncogenomic Changes in PC

### 7.1. Bacterial Metabolites and Their Effect on the Pancreas and PC

Metabolites derived from gut bacteria can affect factors and processes involved in tumorigenesis [37]. Due to the microbial imbalance in PC, bacterial metabolites are largely dysregulated and may act in a pro- or anticarcinogenic way [61]. SCFAs are metabolites mainly stemming from the bacterial fermentation of nondigestible carbohydrates in the colon. They include acetate, butyrate, propionate and lactate, amongst others. Apart from serving as an energy source for colonocytes and other cells, they have an impact on the composition of the microbiome in the colon by regulating the pH and modulating the immune system [37]. In addition, they can modulate epigenetics, gene expression, cell proliferation, and apoptosis [82]. In PC, butyrate was found to have an antiproliferative effect on cultured PDAC cells [83]. In the study by Ren et al., dysbiosis in PC was associated with a decrease in butyrate-producing bacteria [61].

Other bacterial metabolites that are dysregulated in PC are polyamines [84]. Polyamines are polycationic alkylamines that are involved in multiple oncogenic and cell signaling pathways [85]. Since they are essential for cell growth, their absence leads to cell-cycle arrest [86]. A large number of gut bacteria synthesize, accumulate, or utilize polyamines [87]. In a mouse model of PC, serum polyamine levels were significantly elevated and associated with cell proliferation and tumor progression [88]. However, in a study on human PDAC, the transport pathways of some polyamines were upregulated, whereas the biosynthesis and the transport pathways of other polyamines were downregulated in tumor tissue [61]. Polyamines can be utilized in targeted therapy or as biomarkers for the early detection of PC [86,88].

LPS is another bacterial component influencing pancreatic cancer. As part of the outer membrane of Gram-negative bacteria, LPS is released into the surrounding environment both during membrane vesicle trafficking and after destruction of the cell wall [89]. LPS plays an important role in the development of PC since it mediates and maintains Toll-like receptor (TLR)-induced inflammation. Levels of LPS-producing bacteria are found to be increased in PC [61].

### 7.2. Bacterial Metabolites and Their Effect on Cancer Mutations

In addition to influencing the tumorigenic processes, bacterial metabolites can have an impact on cancer mutations. In a recent study using a mouse model of intestinal cancer with gain-of-function mutations in Tpr53, Kadosh et al. observed that mutated p53 functioned as a tumor suppressor in the proximal part of the gut (duodenum and jejunum) by disrupting the Wnt pathway, causing a decrease in cell proliferation [90]. On the contrary, in the distal region of the gut (ileum and colon), it acted in an oncogenic gain-of-function manner by activating Wnt signaling and consequently inducing neoplasia. The reason for this opposing function was found to be related to the gut microbiome. Since bacteria are sparse in the proximal but abundant in the distal region of the intestine [91], the authors suspected that certain gut microbial metabolites caused the switch of the mutant p53 from having a tumor-suppressive role to an oncogenic role. They performed a metabolite screen by treating p53-mutated mouse jejunal organoids with several bacterial metabolites known to be associated with tumorigenesis. They found that the bacterial metabolite gallic acid significantly increased Wnt activity and cell proliferation. Gallic acid, a metabolite of *L. plantarum* and *Bacillus subtilis* in the human gut, was found to reverse the Wnt-suppressive function of mutated p53 by epigenetic mechanisms [90]. This study elegantly showed how a bacterially derived metabolite can influence the effect of cancer mutations. To our knowledge, comparable studies have not been conducted on PC yet. It is, however, likely that similar mechanisms act in pancreatic carcinogenesis.

### 7.3. Dysbiosis and Oncogenomic Changes in PC

One of our objectives for this review was to search for possible associations between the microbiota and oncogenomics. Are cancer mutations caused by certain microbiota patterns, or vice versa, are certain microbiota profiles induced by cancer mutations? In CRC, associations between an altered microbiota profile and CRC driver gene mutations have been reported. Higher abundances of *Herbaspirillum* and *Catenibacterium* were associated with mutations in *NRAS* and *TP53*, respectively, while lower abundances of *Barnesiella* were associated with mutated RAS genes [92]. Moreover, several bacterial species involved in CRC tumorigenesis, such as *Escherichia coli*, *Bacteroides fragilis*, *E. faecalis* and *Campylobacter jejuni* are reported to exert carcinogenic or genotoxic effects (reviewed in [93]).

Comparable studies in PC are scarce and direct associations between dysbiosis and cancer mutations have not been reported to our knowledge. However, certain deductions regarding a possible association can be made from some of the studies. A study investigating the associations of different oncogenomic features of PC with the presence of *Fusobacterium* spp. in tumor tissue, found the presence of *Fusobacterium* in tumors to be linked with a worse prognosis. However, no associations between the tumor *Fusobacterium* status and the molecular features typical for PC, including mutations in *KRAS*, *NRAS*, *BRAF*, and *PIK3CA*, epigenetic changes, and mi-R21, mi-R31, or mi-R143 expression levels, could be found [42].

Studies on bacterial infections linked to PC and their possible mechanistic pathways have suggested, that microbes like *P. gingivalis* may have a carcinogenic impact by regulating miRNA expression and thus influencing important immunologic and cancer-related signaling pathways, which may happen in a far-distanced manner [44]. Moreover, the carcinogenic action of bacteria has been summarized as interference in the cell cycle through the induction of DNA damage, mutations, aberrant cell signaling, immune evasion, inflammation, aberrant miRNA expression, and the induction of epigenetic changes [94]. In this regard, the significance of *H. pylori* and *P. gingivalis* in PC have been emphasized [94]. It was hypothesized that the enzyme peptidyl arginine deiminase, secreted by oral bacteria *P. gingivalis*, *Tannerella forsythia* and *Treponema denticola,* could be responsible for *p53* and *KRAS* point mutations in PC through the degradation of arginine [95]. This has, however, not been experimentally proven. In studies on human PC cell lines and xenografts in mouse models, exposure to *P. gingivalis* increased the tumorigenic behavior of cancer cells. The authors postulated *P. gingivalis* to synergize with mutant *KRAS* or other oncogenic factors and thus promote tumorigenesis [96].

In several studies, mutated *KRAS* has been suspected to influence microbiota or microbiota to influence KRAS function and signaling [17,35]. Pushalkar et al. investigated the influence of the pancreatic microbiota on tumorigenesis in PDAC using a mouse model with mutant *Kras* and *Tp53.* They found that PDAC was associated with a distinct gut and pancreatic microbiota that promotes oncogenesis by immune suppression, and implied, that expression of mutant *Kras* might have an impact on the composition and diversity of the gut and pancreatic microbiota [17]. In a similar study that investigated the impact of the host microbiota on PC tumorigenesis both in a mouse model and in human pancreatic tumor tissue, Thomas et al. observed that the gut microbiota accelerated carcinogenesis in the pancreas. By analyzing transcriptomic changes, they found procarcinogenic genes to be upregulated in the presence of the gut microbiota, while anticarcinogenic pathways were upregulated in the absence of the gut microbiota. However, the carcinogenic effect of the gut microbiota seemed to be independent of the *Kras* mutational status [35]. 

In a study on different subtypes of PDAC, an increased abundance of *Acinetobacter*, *Pseudomonas* and *Sphingopyxis* were associated with *KRAS* signaling, DNA replication, and other PC-related pathways in the basal-like subtype [97]. Moreover, Guo et al. found a correlation between microbial β-diversity and host genetics, indicating that host genetic variation can influence the microbiome composition, which may play a role in PC carcinogenesis. For example, genetic variation linked to a lower antimicrobial immune response (affecting genes involved in interferon-γ-mediated infection-related signaling pathways) could cause dysbiosis and expansion of a pathogenic, cancer-promoting microbiota [97].

Chakladar et al. analyzed the relationship between the intrapancreatic microbiota, immunological changes, and gene expression signatures in 187 PDAC patients [54]. They found that the tumor abundance of 13 bacterial species, mostly belonging to the phylum Proteobacteria and especially Gammaproteobacteria, correlated with a poor prognosis and cancer progression. Smokers and males were especially found to harbor a cancer-promoting microbiota and had a worse prognosis. *A. baumannii* and *Mycoplasma hyopneumoniae* were associated with increased oncogenic gene expression signatures in smokers, and *E. coli* and *M. hyponeumoniae* were linked to CNA [54]. The main findings of the studies reporting possible associations of the microbiota with oncogenomic changes, and the analytical methods used are summarized in Table 1.

## 8. Conclusions

This review aimed to investigate the impact of the microbiota on PC and the associations between dysbiosis and oncogenomic changes. We investigated the interactions between the microbiota and pancreas in health and disease. In the healthy pancreas, antimicrobial peptides secreted by the pancreas control the gut microbiota composition, while SCFAs produced by gut microbes induce the secretion of substances like CRAMP or insulin by the pancreas. In PC, the microbiota is dysbiotic in different parts of the gastrointestinal tract. Oral pathogenic bacteria are translocated via intestinal and other routes to the pancreas, where they cause inflammation, which furthermore develops to cancer. The microbiota also has an influence on drug efficacy during cancer therapy, with the ability to either boost or hamper chemo- or immunotherapies. Metabolites of a dysbiotic microbiota affect tumorigenesis in pro- or anticarcinogenic ways. They may induce and maintain the state of inflammation, have an impact on oncogenic and cell-signaling pathways, drive the carcinogenic transformation of premalignant cells or influence tumor cell proliferation. Bacterial metabolites may also change the function of genes through epigenetic mechanisms, as in the example of gallic acid that switches mutated *TP53* from tumor-suppressive to oncogenic.

No direct association between the microbiota and cancer mutations in PC has been detected so far. However, several connections have been observed or hypothesized. *Fusobacterium* was associated with a worse prognosis in PC, while no associations with oncogenomic features of PC were found. The oral pathogen *P. gingivalis* increased the tumorigenic behavior of PDAC cells by inducing cell proliferation and activating cancer signaling pathways. This was speculated to happen through a synergy between *P. gingivalis* and other oncogenic factors such as mutant *KRAS*. *P. gingivalis* was also hypothesized to advance PDAC by altering miRNA expression and to cause point mutations in *KRAS* and *p53*, amongst others. Procarcinogenic genes were found to be upregulated in the presence of the PDAC microbiota. Furthermore, associations between the PDAC microbiota and KRAS-signaling, and associations between certain bacterial species with CNA were detected. Conversely, looking at the influence of PDAC mutations on microbiota, mutated KRAS was postulated to influence the gut and pancreatic microbiota composition and diversity, and certain host-genetic variations were proposed to cause dysbiosis and lead to cancer development. These results illustrate the complex relationship between the microbiota and PC tumorigenesis. The PDAC microbiota profiles that have been generated so far are partly contradictory, and larger studies would be needed to define the microbial landscape of PDAC more clearly. Moreover, studies combining the profiling of both the microbiota and oncogenomics would help to clarify the relationship between these crucial players in PC tumorigenesis. This could contribute to understanding the larger picture of PC formation and progression. The development of noninvasive screening methods based on the oral or stool microbiota could lead to earlier detection and better surveillance of PC. Additionally, the microbiota could be utilized as a predictive marker of treatment response, and treatment outcomes could be improved by microbial depletion of the gut through antibiotics or manipulation of the gut microbiota through FMT, pre- or probiotics to introduce a beneficial microbiota that promotes drug efficacy.

## Figures and Tables

**Figure 1 ijms-22-12978-f001:**
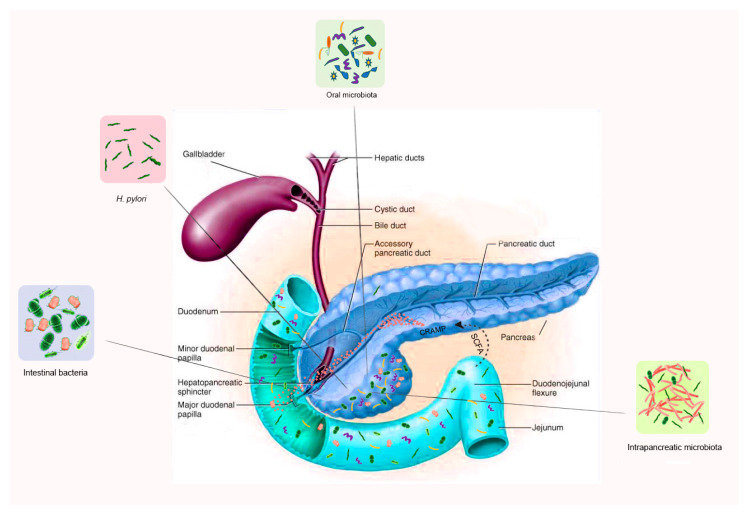
The interplay or cross talk between specific microbiota and pancreatic function, in both health and disease. The short chain fatty acids (SCFA), produced by intestinal bacteria, induce the expression of cathelicidin-related antimicrobial peptides (CRAMP, red dots) in pancreatic β-cells. Impaired CRAMP secretion renders the gut microbiota dysbiotic and results in bacterial overgrowth. The pancreas is colonized under normal physiological conditions by microbes via the pancreatic duct, oral-intestinal route, mesenteric venous drainage, and mesenteric lymphatic drainage.

**Figure 2 ijms-22-12978-f002:**
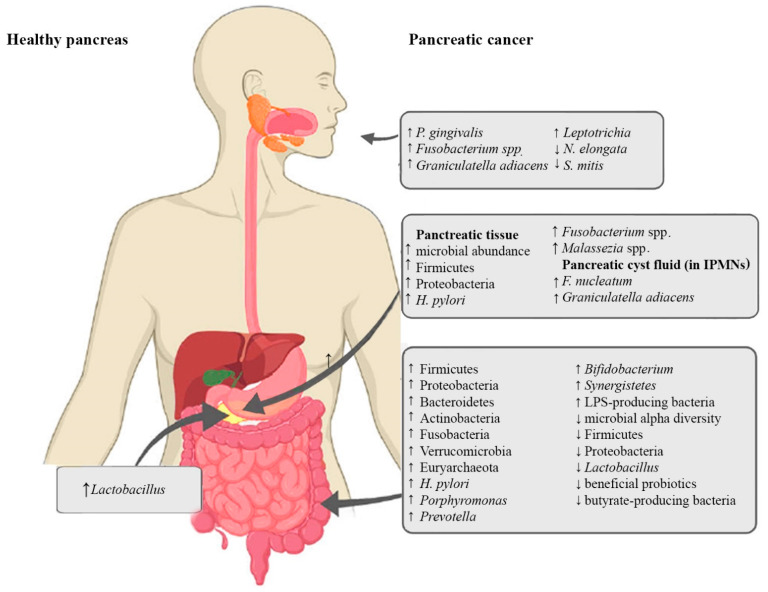
Alterations of oral, pancreatic and intestinal microbiota in PC. ↑ indicates an increase, ↓ indicates a decrease.

**Table 1 ijms-22-12978-t001:** Reviews and studies suggesting associations between microbial and oncogenomic changes in PC.

Reference	Study Population	Specimen	Analytical Methods	Main Findings	Microbial Changes	Oncogenomic Changes	Possible Associations between the Microbiota and Oncogenetics in PC
Mitsuhashi 2015 [42]	Human PDAC vs. HC	Pancreatic	TaqMan Gene Expression Assay	*Fusobacterium* spp. is present in 8.8% of PC tissue and independently associated with a worse prognosis; *F.* spp. could be used as a prognostic biomarker of PC.	↑ *F.* spp. detected in 8.8% of PC tissue specimens.	Mutations in *KRAS, NRAS, BRAF or PIK3CA,* epigenetic changes or mi-R21, mi-R31 or mi-R143 expression levels.	No significant association was found between the *Fusobacterium* species status and molecular alterations of pancreatic cancers.
Michaud 2013 [44]	NA	NA	Review on bacterial infections linked to PC and their possible pathways	Bacteria may cause an inflammatory response of the host immune defense, promoting carcinogenesis.	*H. pylori* and *P. gingivalis* are positively associated with PC.	Aberrant miRNA expression	Microbes like *P. gingivalis* may regulate miRNA expression (even in a far-distance manner), which may influence important immunologic and cancer-related signaling pathways.
Shirazi et al. 2020 [94]	NA	NA	Review aiming to evaluate bacterial agents as cancer biomarkers	Bacteria can influence the cell cycle through inflammation, aberrant cell signaling, immune evasion, DNA damage and mutations, aberrant miRNA expression and epigenetic changes.	*H. pylori* and *P. gingivalis* are associated with PC.	DNA damage, mutations, expression of certain microRNAs, and epigenetic effects	Bacteria involved in carcinogenesis cause alterations in the cell cycle by the induction of DNA damage, mutations, expression of microRNA and epigenetic effects, amongst others. *H. pylori* and *P. gingivalis* cause inflammation and *P.gingivalis* may regulate miRNAs.
Ögrendik 2017 [95]	Human PC	Oral	Hypothesis based on earlier findings	*P. gingivalis, Tannerella forsythia and Treponema denticola* secrete peptidylarginine deaminase, which might cause *p53* and *KRAS* point mutations.	*P. gingivalis,* *T. forsythia* and *T. denticola* are major pathogens of CPO.	Mutations in *p53, KRAS*	Bacterial peptidylarginine deaminases originating from *P. gingivalis, T. forsythia* and *T. denticola* might cause *p53* and *KRAS* point mutations by the degradation of arginine. CPO has been associated with orodigestive cancer.
Gnanasekaran et al. 2020 [96]	Human (PC cell lines, xenograft model)	Pancreatic	Gene expression analysis by qRT-PCR, detection of *P. gingivalis* by RT PCR	Exposure to *P. gingivalis* increases tumorigenic behavior in PC cell lines.	*P. gingivalis* influences PC progression.	Mutant *KRAS*	*P.gingivalis* may synergize with mutant *KRAS* to promote tumorigenesis.
Pushalkar et al. 2018 [17]	Human PDAC, mouse (KPC or KRAS^G12D^ Trp53^R172H^ Pdx^Cre^)	Pancreatic (mouse); pancreatic and fecal (human)	16S rRNA gene sequencing	The PDAC microbiome promotes oncogenesis by immune suppression via TLR; this could be used as a therapeutic target.	↑ Probacteria (*Pseudomonas, Elizabethkingia*) in human PC tissue, is associated with advanced disease; ↑↑ Proteobacteria, Synergistetes, Euryarchaeotain the feces of PC patients.	Mutated *Kras*(G12D)	The composition and diversity of the gut and pancreatic microbiota may be influenced by oncogenic *Kras* expression.
Thomas et al. 2018 [35]	Human PDAC vs. CP and HC; mouse (Kras(G12D)/PTEN^lox/+^)	Pancreatic	16S rRNA gene sequencing, RNAseq of human PDAC xenografts in mice	The pancreatic microbiota in PC accelerates carcinogenesis. No distinct microbiota profile is significantly associated with PC. Gut bacteria exert a long-distance effect on PC carcinogenesis. Bacterial colonization of the pancreas is not a physiological process.	50% of PC mice harbored intrapancreatic bacteria. ↑ *Acinetobacter*, *Afipia, Enterobacter, Pseudomonas* in human PC tissue.	Mutated *Kras* (Kras(G12D)/ PTEN^lox/+^ mouse model)	The gut microbiota accelerates pancreatic carcinogenesis in a mouse model of PC. Many genes involved in carcinogenesis are differently expressed depending on the gut microbiota state. The microbial effect seems to be independent of the *Kras* mutational status. The pancreatic microbiota is not correlated with carcinogenesis.
Riquelme et al. 2019 [53]	Human PDAC STS vs. PDAC LTS	Pancreatic	16S rRNA gene sequencing	Higher α-diversity in the LTS tumor microbiome; predominant taxa could be used as biomarkers for the prediction of LTS; PDAC microbiome composition influences host immune response.	Enrichment of proteobacteria *(Pseudoxanthomanas),* Actinobacteria *(Streptomyces, Saccharopolyspora), Bacillus clausii* in LTS compared to STS.	No genomic differences in PDAC LTS vs. STS.	No genomic differences in stage matched PDAC LTS compared to STS.
Chakladar et al. 2020 [54]	Human PDAC	Pancreatic	Next generation RNA sequencing	The PC tumor microbiota is associated with gene expression dysregulation, metastasis and immune suppression. A worse prognosis in males and smokers is linked to the presence of cancer-promoting microbiota profiles.	*A. ebreus* (Betaproteobacteria) correlated with immune dysregulation and poor prognosis, Gammaproteobacteria correlated with increased metastasis. *A. baumannii* and *M. hypneumoniae* associated with smokers.	Oncogenic gene expression signatures, CNA; Deletions at the 9q13 locus.	An increased abundance of *A. baumannii* and *M. hypneumoniae* is associated with an increase of oncogenic and decrease of tumor suppressive and immune signatures in smokers; *E. coli* abundance is correlated with CNA; *M. hyopneumoniae* is significantly correlated with deletions at 9q13 (potential tumor suppressor) in smokers.
Guo et al. 2021 [97]	Human (different PDAC subtypes)	Pancreatic	Metagenomic sequencing, RNA-seq	Analysis of the tumor microbiome in different subtypes of PDAC: the microbial profile in basal-like PDAC was highly associated with carcinogenesis, possibly through the induction of pathogen-related inflammation. Host genetics influence the composition of the tumor microbiome.	↑ *Acinetobacter, Pseudomonas* and *Sphingopyxis* in basal-like tumors	KRAS signaling	*Acinetobacter, Pseudomonas* and *Sphingopyxis are* associated with carcinogenic gene-expression, KRAS signalling, DNA replication and other PC-related pathways. Bacterial LPS can hyperstimulate KRAS and initiate carcinogenesis. A microbial procarcinogenic effect is caused by continuous inflammation rather than direct mutagenesis. Host genetics participate in shaping the tumor microbiome.

↑ = increased abundance of bacteria in PC compared to HC, ↑↑ = significantly increased abundance of bacteria in PC compared to HC, CNA = copy number alteration, CP = chronic pancreatitis, CPO = chronic periodontitis, HC = healthy controls, LTS = long-term survivors, NA = not applicable, PDAC = pancreatic ductal adenocarcinoma, PC = pancreatic cancer, STS = short-term survivors.

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
