# Peer review of "Microbiota Alterations and Their Association with Oncogenomic Changes in Pancreatic Cancer Patients"

_ijms, 2021, doi:10.3390/ijms222312978_

Round 1
Reviewer 1 Report
The current manuscript reviews the interaction between pancreatic carcinogenesis and microbiota, as well as the role of microbiota in drug efficacy. This is an interesting and timely topic, and the manuscript is, in general, well written. The contemporary literature is cited and discussed.
The authors might want to consider including a graphical overview of the microbial profile in pancreatic health and disease.
Author Response
We thank the reviewer for their positive comments. As recommended, we have now made and added a new Figure 2 to represent a graphical overview of the microbial profile in pancreatic cancer.
Reviewer 2 Report
The manuscript by Sammallah et al. “Microbiota Alterations and their Association with Oncogenomic Changes in Pancreatic Cancer Patients” is interesting. There are a few concerns to be addressed before its publication as follows:
Comments.
- Overall, the author should provide more useful quantities of data on actual microbial population changes association to pancreatic cancer patients like how much population changes of different organisms in microbiota, and how they influence health, etc.
- Line 77-81, the authors may elaborate the information with a few microbe’s likes Clostridium and Salmonella, etc in cancer research and health. Doi: 10.1016/j.semcancer.2021.05.012.
- Section 4, please provide a few quantitative data of microbial population changes.
- Table 1, Microbial changes entries may be more elaborated with detailed quantitative data.
Author Response
- We value the reviewer’s comment and have tried to add as much quantitative information as available from the reviewed studies. All additions are marked with tract changes in the manuscript. However, since not all studies include enough quantitative data, we have added words like ’significant’ to refer to statistically significant differences in bacterial abundances.
-
As suggested by the reviewer, we have now added more detail information, including more specific examples of Clostridium and Salmonella in cancer research. The text included is marked with tract changes in the manuscript, lines 81-85.
- In response to reviewer’s suggestion, we have now added more text mentioning quantitative changes in microbial population and how these influence health.
- We have now improved our table based on the recommendation by the reviewer and have added some quantitive data where available. We have added words like ’significant’ to refer to statistically significant differences in bacterial abundances.